# Usefulness of COL11A1 as a Prognostic Marker of Tumor Infiltration

**DOI:** 10.3390/biomedicines11092496

**Published:** 2023-09-08

**Authors:** Javier Freire, Pilar García-Berbel, Belén Caramelo, Lucía García-Berbel, Victor J. Ovejero, Nuria Cadenas, Ainara Azueta, Javier Gómez-Román

**Affiliations:** 1Pathology Department, University Hospital Marques de Valdecilla, Avda. Marqués de Valdecilla s/n, 39008 Santander, Spain; 2Pathology and Molecular Pathology Unit, IDIVAL, Avenida Cardenal Herrera Oria s/n, 39011 Santander, Spain; 3Breast Unit, Gynecology Department, University Hospital Puerta del Mar. Av. Ana de Viya, 21, 11009 Cádiz, Spain; 4Surgery Department, University Hospital Marques de Valdecilla, Avda. Marqués de Valdecilla s/n, 39008 Santander, Spain; 5El Alisal Health Center, Cantabrian Health Service, C. los Ciruelos, 48, 39011 Santander, Spain

**Keywords:** COL11A1, collagen XI alpha 1, tumor metastasis, tumor activated fibroblast

## Abstract

Background: Determining the infiltration of carcinomas is essential for the proper follow-up and treatment of cancer patients. However, it continues to be a diagnostic challenge for pathologists in multiple types of tumors. In previous studies (carried out in surgical specimens), the protein COL11A1 has been postulated as an infiltration marker mainly expressed in the extracellular matrix (ECM). We hypothesized that a differential expression of COL11A1 may exist in the peritumoral stroma of tumors that have acquired infiltrating properties and that it may be detected in the small biopsies usually available in normal clinical practice. Material and methods: In our study, we performed immunohistochemical staining in more than 350 invasive and noninvasive small samples obtained via core needle biopsy (CNB), colonoscopy, or transurethral resection of bladder tumor (TURBT) of breast, colorectal, bladder, and ovarian cancer. Results: Our results revealed that COL11A1 immunostaining had a sensitivity to classify the samples into infiltrative vs. noninfiltrative tumors of 94% (breast), 97% (colorectal), >90% (bladder), and 74% (ovarian); and a specificity of 97% (breast), 100% (colorectal), and >90% (bladder). In ovarian cancer, the negative predictive value (0.59) did not present improvement over the usual histopathological markers. In all samples tested, the cumulative sensitivity was 86% and the specificity 96% (*p* < 0.0001). Conclusions: COL11A1-positive immunostaining in small biopsies of breast, colon, bladder and ovarian cancer is an accurate predictive marker of tumor infiltration that can be easily implemented in daily clinical practice.

## 1. Introduction

Cancer is undoubtedly a global public health problem and, despite the efforts made, currently maintains high morbidity and mortality rates [1].

Although the cellular processes that occur during carcinogenesis can promote the appearance and growth of the tumor, it is the way of dissemination and the metastasis that ultimately cause the death of the patient [2,3,4]. Both clinicopathological stages and medical treatments of cancer are mainly based on the presence or absence of distant metastasis, so the study of this capacity seems to be a priority in the knowledge of cancer. We know that during the epithelial–mesenchymal transition (EMT), phenotypic changes in the epithelial cell may occur: intercellular adhesion alterations, cell interactions with the environment, changes in the cytoskeleton, degradation of the basement membrane, and rearrangement of the extracellular matrix (ECM) [5]. This matrix is composed of numerous proteins that contribute to its structure and function such as collagens, fibronectin, laminins, glycosaminoglycans, proteoglycans, and remodeling enzymes [6], with the collagens being the main components of the ECM and representing about 30% of the total protein mass [7]. During the process of tumor invasiveness, significant changes are observed in the collagen composition of the ECM, highlighting an accumulation of fibrillar collagens compared with a collagen type IV decrease [8]. Within all types of collagen, in recent years, collagen XI has been proposed as a priority driver of tumor invasiveness [9,10,11].

COL11A1 was first described in oncology in a rhabdomyosarcoma cell line [12]. Since then, it has been studied in many tumors not only in epithelial-origin tumors such as lung, breast [13], pancreas [14], gastric [15], or colorectal [16,17] tumors, where it seems to be overexpressed compared with normal epithelium or inflammatory lesions, but also in mesenchymal-derived pathology such as angiosarcoma [18], osteosarcoma [19], or sarcoma of the extremities [20], where the presence of COL11A1 appears to be a negative prognostic factor. All these previous studies support the idea that the expression of COL11A1 is a particular event in the remodeling of the ECM that confers invasive capacity; therefore, its use as a diagnostic infiltration marker could be advantageous.

Our hypothesis is that there is a differential expression of COL11A1 in the peritumoral stroma of tumors that have acquired infiltrating properties compared with noninfiltrating neoplastic lesions. Furthermore, since it is possible to analyze the expression of this protein via immunohistochemistry, it could be used as an infiltration-positive marker in the daily practice of a pathology service.

## 2. Materials and Methods

We performed a literature review to determine the most common problems that appear in the differential diagnosis of minimally invasive and preneoplastic lesions in the pathologist’s daily practice. We then selected different breast, colon, bladder, and ovarian cancer samples to test the utility of COL11A1 immunohistochemistry in infiltration diagnosis.

In order to mimic clinical practice, small diagnostic biopsy samples were selected: core needle biopsy (CNB), endoscopic biopsies, and transurethral resection of bladder tumor (TURBT).

### 2.1. Breast Cancer

Two hundred one patients with different breast infiltrating carcinomas without diagnostic difficulties were selected to assess the sensitivity of the technique in a controlled environment. Subsequently, to validate the technique for its medical application, a second population composed of 40 in situ carcinomas that presented considerable diagnostic difficulties in CNB was selected. Out of these, 21 samples presented infiltration in the subsequent surgical resection.

### 2.2. Colorectal Cancer

As with breast cancer, in colon cancer, we followed a double approach; on the one hand, we studied COL11A1 expression on 30 adenocarcinomas compared with noninvasive tumor lesions (polyps) or non-neoplastic lesions (21).

Afterward, complex cases that had generated doubts in the diagnosis were searched. The expression profile of COL11A1 was determined in 23 in situ carcinomas settled on tubular adenomas (ISCSTA), of which 13 had tumor infiltration in the subsequent surgery. These last samples were obtained via colonoscopy in order to determine the marker predictive value in real struggling conditions.

### 2.3. Bladder Cancer

In order to accurately differentiate the worst-prognosis lesions (pT2) from those with a better outcome (pTa and pT1), the expression of COL11A1 was determined in TURBT samples in which diagnosis was challenging due to their small size and low-quality tissue as a consequence of the artifacts generated by electrical burns during surgery. Forty-three samples were selected as follows: eleven pTa, fifteen pT1, eleven pT2, and six non-tumoral lesions.

### 2.4. Ovarian Cancer

To determine the potential of COL11A1 in the detection of infiltration in ovarian cancer, the differential expression of this marker was compared in 27 borderline-diagnosed carcinomas with subsequent infiltration with respect to 10 noninvasive lesions.

### 2.5. COL11A1 IHQ Protocol

Formalin-fixed, paraffin-embedded samples were stained using proCol11a1 monoclonal antibody clone 1E8.33 (ONCOMATRYX, Bilbao, Spain). A Roche-Ventana Ultra Benchmark automatized system (Roche, Oro Valley, AZ, USA) was used to perform immunohistochemistry. Antigen retrieval was performed with Ultra CC2 buffer for 16 min at 98 °C; antibody was diluted at 25 µg/mL and incubated for 30 min at RT. OptiView kit (Roche-Ventana) was used to stain the samples, according to the manufacturer’s protocol. Staining was separately evaluated by two independent pathologists. COL11A1 was considered positive when at least one cancer-associated fibroblast (CAF) presented clear cytoplasmic immunostaining as described previously [21].

### 2.6. Statistics Analysis

Microsoft Excel 2013 ((New México, USA)) was used to manage all the clinical and histopathological parameters. Statistics were calculated using SPSS (version 20) packages and GraphPad Prism (version 6.01). The analysis of the differential expression of COL11A1 was carried out using a chi-squared test with Fisher’s correction. In all cases, *p* < 0.05 was considered as significant.

## 3. Results

### 3.1. Breast Cancer

COL11A1 expression was observed as cytoplasmic staining in fibroblasts surrounding the tumor. Ninety-four percent of infiltrating carcinomas were positive (99 out of 105) while only 3 of noninvasive samples showed staining (Figure 1a,b).

Out of the 40 in situ selected carcinomas, 21 presented some microinfiltration areas in the surgical proceedings. Serial sections were created to compare the expression of myoepithelial cell markers used in diagnostic routine (p63 and Calponin) with the expression of COL11A1. Among the 21 microinfiltrative samples cataloged as in situ using myoepithelial cells markers, 19 would have been positive for infiltration with the application of COL11A1 (Figure 1c,d). The expression of pro-COL11A1 presents, therefore, a significantly higher capacity to determine the microinfiltration (*p* < 0.0001) than calponin or p63, with a sensitivity of 94% and a specificity of 97% when used as a marker of infiltration in breast tumors. In addition, the expression of pro-COL11A1 in breast lesions presents a likelihood ratio of 14, a positive predictive value of 0.97, and a negative predictive value of 0.93. Note that pro-COL11A1 is a positive marker, something that is easier to detect than an absence of staining of p63 or calponin, which may be focal.

### 3.2. Colorectal Cancer

As in breast cancer, the aim of this work in colon cancer was to determine the diagnostic utility of COL11A1 in small samples where microinfiltration is difficult to identify.

In a first approach, the COL11A1 expression of typical adenocarcinomas was compared with that of noninfiltrative lesions in order to determine the validity of this marker. From 30 adenocarcinomas, 29 presented COL11A1 immunolabeling, while 0 of the 21 noninvasive samples presented positive expression (*p* < 0.0001). These data demonstrate a sensitivity of 97% and a specificity of 100%. As in breast tumors, staining was observed in CAFs (Figure 2).

Otherwise, COL11A1 expression was determined in the lesion that implies a greater diagnostic challenge, that is, in situ carcinomas settled on adenomatous polyps. For this, 23 of these lesions were selected with and without infiltration in the subsequent surgery. Out of the 13 lesions that presented infiltration in the surgical specimen, 8 were immunopositive for the expression of COL11A1, while none of the pure in situ lesions presented expression (*p* = 0.0027). Staining was observed in the fibroblastic nests surrounding the tumor (Figure 3). With a sensitivity above 60% and a specificity of 100%, this marker could be a complementary diagnostic tool to determine microinfiltration in colonic lesions.

### 3.3. Bladder Cancer

Differentiation of poor prognosis staging (pT2) from those with better prognosis (pTa, pT1) in RTU sometimes present huge difficulties, mainly due to intrinsic technique complications; COL11A1 was tested to determine if the expression of this marker is associated with muscular infiltrating carcinomas.

Of the total pT2-stage samples, 91% (10 of 11) presented positive immunostaining, while 31 of the 32 noninfiltrating samples were negative for marker expression. As in other tumors, strong cytoplasmic staining was observed in the CAFs of pT2 samples (Figure 4).

These results present a significant differential expression (*p* < 0.001) between the biopsies diagnosed as pT2 and those that do not present invasion of the *Muscularis propria* (MP)*,* with a sensitivity and specificity greater than 90%.

### 3.4. Ovarian Cancer

This tumor location was selected to determine the clinical usefulness of COL11A1 in predicting invasiveness in borderline ovarian carcinomas, which pose serious difficulties in accurate diagnosis.

Of the total of the invasive lesions, 17 presented immunostaining for COL11A1 while 0 of the 10 nontumor lesions presented immunolabeling (Figure 5). Although the differential expression between infiltrating and noninvasive lesions was significant (*p* < 0.005), both the sensitivity (74%) and, mainly, the negative predictive value (0.59) do not present improvement over those of the current examination method.

## 4. Discussion

### 4.1. Collagen XI Alpha 1 as a Tumor Infiltration Marker

It was described in a meta-analysis that overexpression of COL11A1 (measured using mRNA) is a differential event between high- and low-grade tumors in different carcinoma phenotypes [22]. This meta-analysis also demonstrated that COL11A1 expression was found among genes with the highest differential expression in ovarian, breast, colon, and lung tumors ^18^. In addition, there are many studies based on RNA expression profiles that propose that the overexpression of this protein may play an important role as a differentiating marker between invasive and noninvasive tumor lesions [14,16,23,24,25]. The present work contributes in a significant way to supporting this hypothesis, generating data that demonstrate the diagnostic utility of this biomarker.

The use of samples based on minimally invasive biopsies (CNB, colonoscopy, and TURBT) presents a diagnostic approach much closer to real practice than the methods published up to date, where the study of expression has been carried out with surgical specimens [22].

To perform this work, an extremely specific anti-COL11A1 [26] antibody was used. COL11A1 has a high homology with COL5A1, not only structurally but also functionally [27]; both collagens 5 and 11 regulate the diameter of the two main type I and II collagens (respectively) [28]. Therefore, the use of an antibody against a highly specific region of COL11A1 prevents cross-reactivity with COL5A1, which could generate false positives. In addition, the use of a monoclonal antibody improves the specificity, and the use of the immature form of collagen (procollagen) provides a pattern of intracellular cytoplasmic expression, which is much easier to analyze with respect to extracellular immunostaining.

It should be noted that for those tumors where the COL11A1 usefulness has been demonstrated, the expression of this marker can predict invasiveness with a very high sensitivity and specificity. In fact, in analyzing the differential expression in the different tumor samples classified as invasive versus noninfiltrative, we have found a cumulative sensitivity of 88% and a specificity of 97% in a population of over 350 samples (*p* < 0.0001) (Table 1).

It seems, therefore, that we have an extremely precise infiltration marker and that, in addition, it can be used in the daily clinical analysis of multiple tumor types.

### 4.2. Breast Cancer

The correct diagnosis of microinfiltrative lesions in breast carcinomas is, in many cases, a serious challenge for the pathologist; in fact, many articles have determined the existence of a large diagnostic underestimation in the malignant potential of noninfiltrative lesions [29,30,31].

To determine tumor microinfiltration, pathologists have been using various markers over the years such as the loss of the basal membrane [32,33,34], the S-100 protein, or the high-molecular-weight cytokeratins [35,36]. Currently, the absence of p63 or calponin is the gold standard to determine tumor infiltration; however, its sensitivity is controversial. Since p63 is a nuclear marker, the absence of signal may be due only to the histological section causing false positives for infiltration. Likewise, the rapid metabolism of calponin can hinder its expression and therefore generate a negative result, thus incurring a false positive for infiltration [37].

The use of COL11A1 as a tumor invasiveness marker presents some advantages over other markers. Firstly, and most important, and as confirmed by our data, the diagnostic sensitivity of COL11A1 is much greater than that of calponin and p63. The cases of in situ carcinomas with subsequent infiltration were selected with samples in which loss-of-myoepithelial-cells-based tools had failed. In addition, performing the techniques in consecutive sections substantially increases the reliability of the experiment, since the invasiveness in the same area can be determined. Secondly, as COL11A1 is a positive marker, it greatly facilitates the interpretation and therefore diagnosis, since it can often be difficult to evaluate loss of immunostaining [32]. And lastly, as COL11A1 is a stromal marker (directly related to infiltration), its use represents a breakthrough, especially in small samples (e.g., CNB) since, unlike markers of myoepithelial cell loss that require infiltration area to be present [38,39], COL11A1 is able to detect stromal changes regardless of whether there is an infiltrative tumor component or not.

### 4.3. Colorectal Adenocarcinoma

Information on the histopathological characteristics of colonic adenomas is critical for its management, since microinfiltrating lesions and high-risk and advanced neoplasms require completely different strategies of treatment and follow-up. Pathologically, advanced lesions are defined by size and morphology, so a valid tissue sampling is crucial for therapy election and surveillance recommendations [40].

Although colonoscopy is considered to be the best tool for the screening of colorectal cancer [41], several studies have concluded that diagnosis based on endoscopic biopsies underestimated the histopathological diagnosis in approximately 10% of colorectal adenomas [40]. For this reason, new tools are required to predict the malignancy of those 10% cases without accurate diagnosis.

COL11A1 overexpression, measured using mRNA, was first referenced in sporadic colonic carcinomas compared with normal colon tissue of colorectal cancer [17]. After this, there have been several publications that have described the overexpression of COL11A1 in tumor samples when compared with noninvasive samples or normal tissue [42,43,44]. However, until now, its use as a predictive biomarker of infiltration in a clinical diagnosis context has not been proposed.

This work presents the usefulness of COL11A1 in the differential diagnosis of infiltration in colon cancer. To date, there are no diagnostic tools to help with the diagnosis of complex colonic lesions, such as the microinfiltration of carcinomas seated on adenomatous polyps [45].

This work demonstrates that COL11A1 expression is associated with an increased risk of infiltration. Along the same line, and surprisingly, COL11A1 seems to be also overexpressed in colon polyps of patients with adenomatous familial polyposis via mutation in APC [16]. As the clinical treatment of these polyps is their resection, we suggest COL11A1 expression determination in all polyps would be very interesting in order to perform an early detection of microinfiltration or malignancy predisposition [46].

Several papers have presented other prognostic biomarkers, such as the mutation of exon 54 of the COL11A1115 gene or a malignant genes signature [47]. However, although its clinical usefulness was not discussed, it is technically much easier to perform a single immunohistochemistry, which provides the same result.

### 4.4. Bladder Cancer

AJCC cancer stage is the most important prognostic factor in urothelial carcinoma [48]. When the invasion of the MP is identified (pT2), aggressive treatment is usually required (cystectomy, systemic chemotherapy, or radiotherapy). Management of T1-stage tumors is more conservative, although the optimal procedure for these tumors is still controversial [49]. Consequently, the precise staging in TURBT is crucial for the proper management of bladder tumors.

The main problem found in TURBT samples is the differentiation of the muscular mucosa (MM) of the MP when MM hyperplasic phenomena or due destruction by tumor invasion occur [50,51]. In addition, intrinsic problems of the technique, such as limited specimens, lack of orientation, or cauterization artifacts, make accurate microinfiltration diagnosis challenging [49,52].

Recently, some infiltration biomarkers’ proteins have been described. Specifically, smoothelin, a cytoskeleton protein that is expressed specifically in the contractile smooth muscle, seems to have a strong expression in MP, whereas MM does not present it or the expression is very weak [53,54,55]. However, there are articles advocating careful use, since there seems to be an overlapping immunostaining pattern of up to 25% in MM compared with MP [49].

In the last few years, several works have reported COL11A1 as a tumor promoter in transitional bladder carcinoma relationships [56,57]. Similar to our results, COL11A1 appears as one of the seven genes differentially expressed when comparing MM invasion against MP [58]. Our work provides several advantages over previous studies: the single expression of COL11A1 is able to predict MP infiltration, and it is a nondependent MP presence marker (as smoothelin) because it is tumor stromal marker. Additionally, it presents very high sensitivity and specificity ranges.

### 4.5. Ovarian Cancer

Ovarian borderline tumors prognosis differs greatly from that of infiltrative carcinoma. Unlike patients with infiltrating ovarian carcinomas, the vast majority of patients with borderline tumors show excellent survival rates [59]. The 5-year survival rate for borderline stage I tumors is approximately 95% to 97%, and even patients with borderline tumors in stages II or III have a 5-year survival rate of more than 65% [60].

The correct diagnosis of borderline tumors is complex. These tumors share morphological characteristics with malignant infiltrating tumors, which may confuse the pathologist [61,62]. Previous studies determined an underdiagnosis of between 20 and 30% in borderline tumors [63,64], mainly in tumor stages III or IV [65].

There have been various biomarkers proposed to determine the malignant potential of borderline tumors, such as differential loss of p14 [66] in invasive tumors with respect to borderline, overexpression of interleukin 8 (IL-8) and its receptors [67], or multigenic signature to predict worse prognosis [68]; however, none of them have demonstrated high sensitivity rates.

Wu et al. demonstrated that COL11A1 overexpression was significant on ovarian tumors and that the expression level correlated with tumor stage [69]. Subsequently, other studies supported these data and determined not only that there is an overexpression of COL11A1 in invasive tumors but also that overexpression is associated with a worse prognosis and a higher rate of recurrence [70,71]. However, there has not been any published work about its clinical diagnostic application or differentiation in borderline tumors.

Our data suggest that although COL11A1 seems to predict malignancy, consistent with findings published to date, contrasting with other tumor types presented in this work, it does not have a sufficient sensitivity for its use in diagnostic routine. According to the data, almost 30% of the infiltrating tumors could be underestimated, so it does not seem to improve the sensitivity compared with that of conventional anatomopathological analysis. It would be interesting, however, to determine the value of the combination of several previously mentioned markers (p14 or IL-8) with COL11A1 to see if a higher diagnostic sensitivity is achieved.

The present work shows how COL11A1 expression is a solid predictive marker of tumor infiltration. It was demonstrated that the positive immunostaining for this protein in CAFs of breast, colon, bladder, and ovarian cancer predicts neoplastic invasion with a high sensitivity and specificity. These data suggest that COL11A1 expression would allow therapeutic decisions to be made in situations of diagnostic uncertainty.

## Figures and Tables

**Figure 1 biomedicines-11-02496-f001:**
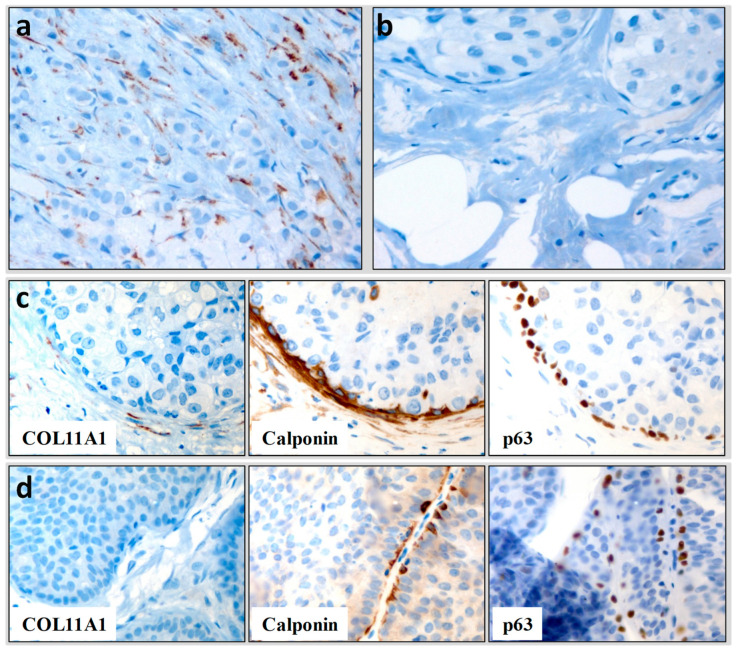
Differential COL11A1 expression in breast lesions. COL11A1 immunostaining (**a**) invasive ductal carcinoma, (**b**) in situ ductal carcinoma, (**c**) microinvasive ductal carcinoma for both COL11A1 and myoepithelial cell markers, and (**d**) in situ ductal carcinoma for both COL11A1 and myoepithelial cell markers. Hematoxylin and eosin, HE. Original magnification ×360.

**Figure 2 biomedicines-11-02496-f002:**
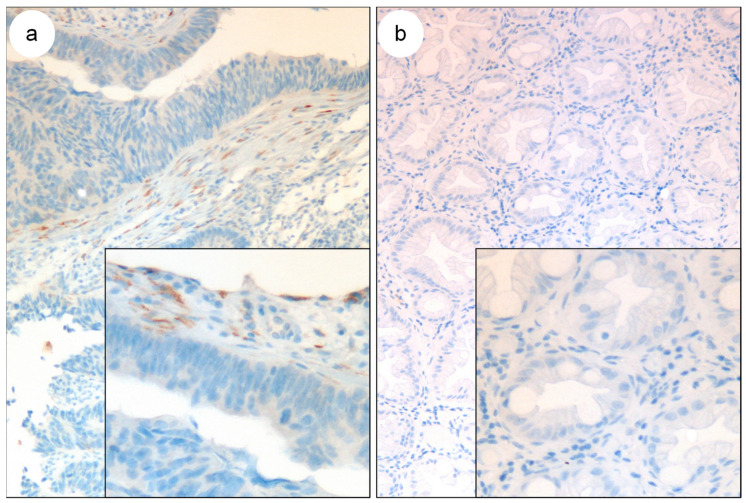
COL11A1 expression in colonic neoplasia. (**a**) Colonic adenocarcinoma; (**b**) hyperplastic polyp. HE. Original magnification ×180, box ×360.

**Figure 3 biomedicines-11-02496-f003:**
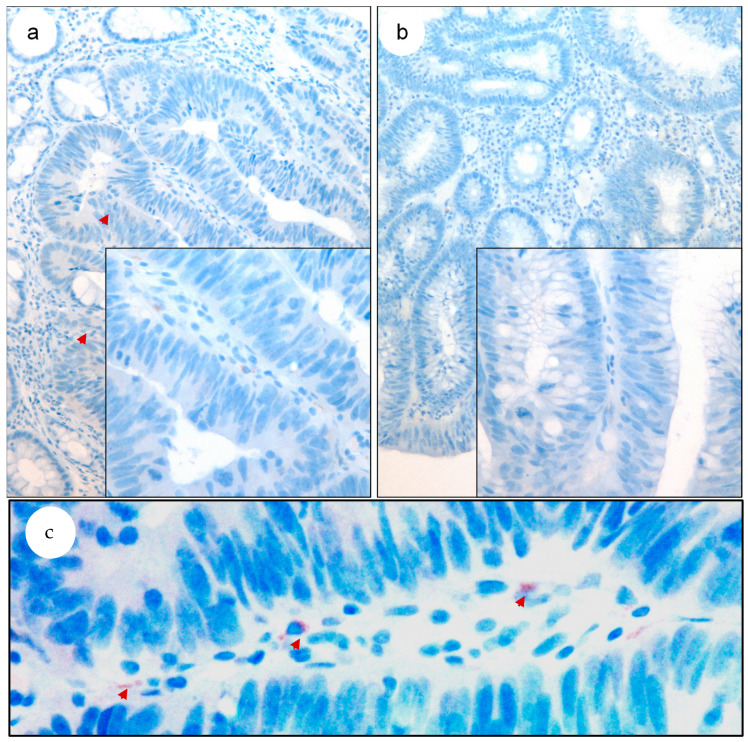
COL11A1 expression in colonic in situ carcinomas settled on adenomatous polyp. (**a**) With subsequent infiltration; (**b**) pure in-situ. HE. (**c**) Amplification of the immunopositive region of box A. Original magnification ×180, box small ×360, big ×600.

**Figure 4 biomedicines-11-02496-f004:**
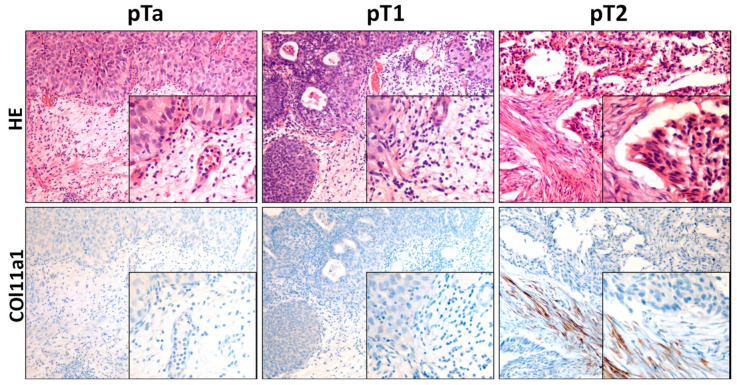
Immunostaining for COL11A1 in transurethral biopsy of urinary bladder. Immunostaining is observed in pT2 samples, while none of pTa or pT1 show expression. HE. Original magnification ×180, box ×360.

**Figure 5 biomedicines-11-02496-f005:**
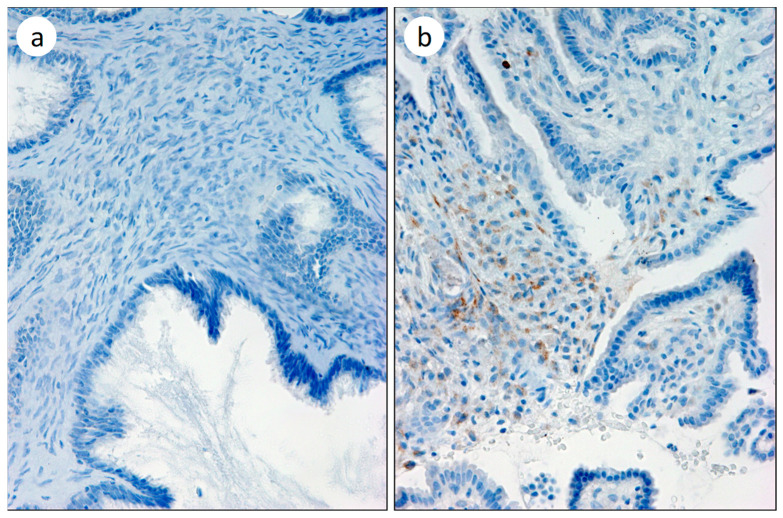
COL11A1 Immunostaining in ovarian carcinoma. (**a**) Negative mucinous cistoadenocarcinoma; (**b**) positive papilar carcinoma.

**Table 1 biomedicines-11-02496-t001:** Summarized results for each cancer type.

	Infiltrative		Noninfiltrative			
	Positive COL11A1 IHQ	Negative COL11A1 IHQ	Positive COL11A1 IHQ	Negative COL11A1 IHQ	Sensitivity	Specificity
Breast	118	8	5	110	0.94	0.96
Colorectal	37	6	0	31	0.86	1.00
Bladder	10	1	1	31	0.91	0.97
Ovarian	17	10	0	10	0.63	1.00
Overall	182	25	6	182	0.88	0.97

## Data Availability

Not applicable.

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
