# Peer review of "Usefulness of COL11A1 as a Prognostic Marker of Tumor Infiltration"

_biomedicines, 2023, doi:10.3390/biomedicines11092496_

Round 1
Reviewer 1 Report
This article focuses on the expression of COL11A1 in tumor stromal tissue and proposes that its expression in paraneoplastic regions is a certain indicator of tumor invasion there.
The study examined 201 breast cancer cases, 30 colorectal cancer cases, 43 bladder cancer cases, 27 ovarian cancer cases, in.
They reported that when COL11A1 was positive in stromal cells, the presence of tumor invasion was 94%, 97%, >90%, and 74% in each carcinoma.
The point of view seems to be good, but as for the analysis, in the end, is the check for the presence of invasion a histopathological observation? (How was the tumor tissue side identified?).
It would be interesting to know the molecular mechanism of the role of COL11A1 in tumor tissues if this finding is the trigger, but this paper only captures the phenomenon at first and does not pursue it in depth.
Author Response
Point 1: The point of view seems to be good, but as for the analysis, in the end, is the check for the presence of invasion a histopathological observation? (How was the tumor tissue side identified?).
Response 1:
Thank you for the question. In the text, the Materials and Methods section outlines the methodology employed to assess infiltration. In conventional clinical practice, the initial diagnostic procedure tends to be minimally invasive (such as fine-needle biopsy, core needle biopsy, or colonoscopy), yielding a limited amount of tissue that is sometimes not entirely representative of the tumor. Hence, the true diagnostic reality is established through analysis of the specimen post-surgery, where larger and more representative samples of the lesion are obtained. The sample selection was conducted on those cases where the small biopsy posed a diagnostic challenge due to lack of correlation (in terms of infiltration) with the surgical specimen.
Point 2: It would be interesting to know the molecular mechanism of the role of COL11A1 in tumor tissues if this finding is the trigger, but this paper only captures the phenomenon at first and does not pursue it in depth
Response 2: Certainly, understanding the underlying biology of COL11A1 expression would constitute a significant advancement, potentially serving as a marker to guide future therapeutic approaches. However, the present study does not aim to delve into the underlying pathways governing the expression of this protein. Rather, we highlight a highly valid diagnostic tool that outperforms the currently employed methods. The ultimate objective of this research is to furnish the practicing pathologist with a novel target that facilitates swifter and notably more accurate diagnoses.
Reviewer 2 Report
The authors of this manuscript investigated the role of COL11A1 protein as prognostic biomarker for tumor infiltration in different cancer types including breast, colorectal, bladder and ovarian tumors. To assess their hypothesis, they performed IHC staining of more than 350 tumor samples and reported a great sensitivity and specificity of COL11A1 in discriminating infiltrative vs non-infiltrative tumors, particularly for breast and colorectal cancers.
The topic is of interest since it focuses on a potential easy-to-use prognostic marker able to predict infiltrative behaviour of cancers for which can be challenging due to limited amount of tissues available from biopsies.
However, some minor suggestions are provided in order to further improve the overall quality of the manuscript:
1) the involvement of collagen in cancer development is widely studied, with collagen XI being known for its role as driver in promoting invasiveness in many different cancer types. Besides all the other types mentioned by the authors, COL11A1 has been related also with with mesenchymal tumors such as sarcomas, in particular rhabdomyosarcoma (as mentioned) but also angiosarcoma, osteosarcoma and also sarcoma of the extremities such as UPS. Indeed, in a recent study in which RNAseq analysis was performed comparing myxofibrosarcoma vs undifferentiated pleomorphic sarcoma patients, COL11A1 was found significantly up-regulated in UPS cases compared to MFS ones. This is in line with clinical–pathological features of UPS as an hystotype prone to distant recurrence and metastasis. Please discuss these findings in Introduction section adding the following relevant references: doi.org/10.3390/ijms24086926, doi.org/10.1186/s12967-019-2058-1, doi: 10.3390/cancers13050935, doi: 10.1111/cas.14726, doi: 10.1177/1535370217736512.
2) Information about approval of the study from the Ethics Committee as well as written informed consent from the patients should be provided
3) a table summarizing number of positive/negative samples for each cancer type with related percentages and main features (presence/absence of microinfiltration areas) would be helpful to get a quick overwiew about main findings
Overall English language is fine, with only minor grammar/spelling mistakes and few incomplete sentences. Minor editing of the text is required.
Author Response
The questions have been answered within the article's text (highlighted in red) . Additionally, a summary table has been appended to the discussion section to aid in the article's readability. Minor textual modifications have been made to ensure coherence.

Round 2
Reviewer 1 Report
Now, authors alteration of manuscript in R1 would be fine.